# Advance Directives and Factors Associated with the Completion in Patients with Heart Failure

**DOI:** 10.3390/ijerph18041780

**Published:** 2021-02-12

**Authors:** JinShil Kim, Mi-Seung Shin, Albert Youngwoo Jang, Shinmi Kim, Seongkum Heo, EunSeok Cha, Minjeong An

**Affiliations:** 1College of Nursing, Gachon University, 191 Hambakmeoro, Yeonsu-gu, Incheon 21936, Korea; kimj503@gachon.ac.kr; 2Division of Cardiology, Gil Medical Center, Department of Internal Medicine, Gachon University College of Medicine, 21 Namdong-daero 774 beon-gil, Namdong-gu, Incheon 21565, Korea; albert.jang.md@gmail.com; 3Department of Nursing, Changwon National University, 20 Changwondaehakro, Euichanggu, Kyungsangnamdo, Changwon 51140, Korea; skim@changwon.ac.kr; 4Georgia Baptist College of Nursing, Mercer University, 3001 Mercer University Drive, Atlanta, GA 30341, USA; heo_s@mercer.edu; 5College of Nursing, Chungnam National University, 266 MunWharo, Junggu, Daejeon 35015, Korea; echa5@cnu.ac.kr; 6College of Nursing, Interdisciplinary Program of Arts & Design Technology, Chonnam National University, 160 Baekseoro, Donggu, Gwangju 61469, Korea

**Keywords:** heart failure, advance directive, prognosis, advance care planning, palliative care

## Abstract

Advance directive (AD) has been underutilized among patients with heart failure (HF). This study was performed to explore the ADs and examine factors associated with the completion of an AD survey in patients with HF. In a descriptive, correlational study, data on end-of-life values, treatment directives, and proxy (Korean-Advance Directive (K-AD) questionnaire) and factors associated with K-AD completion were collected among HF patients during outpatient visits. Of 67 patients (age, 67 years; male, 61.2%), 52.2% completed all or part of the K-AD. Among values, comfortable death was the most preferred (*n* = 15) followed by avoiding family burden (*n* = 6). In those completers, preferences for hospice care, cardiopulmonary resuscitation, ventilation support, and hemodialysis were 68.6%, 42.9%, 28.6%, and 28.6%, respectively. Female sex (odds ratio (OR) = 0.167), poorer HF prognosis (OR = 0.156), and better functional status (OR = 0.905) were associated with less likelihood of completing the AD survey. The findings suggest that in-depth AD discussion needs to be started earlier in patients with HF to facilitate completion of AD, especially in female patients. Future research should investigate if early discussion of ADs as part of advance care planning with integration into standard care of HF facilitates the documentation of ADs.

## 1. Introduction

Heart failure (HF) is considered a global pandemic health concern in Western and Asian countries today [1,2,3]. Approximately 26 million people across the world have HF [3]. Patients with HF live longer due to therapeutic advances [4,5], but experience high rates of cardiac or non-cardiac morbidity and mortality during the progressive debilitating trajectory of illness, which lead to experiencing escalated physical, psychological, and financial burdens [3].

Detrimental burdens of HF demand global efforts to improve the quality of HF management. Patients and/or their families face a wide spectrum of decision-making related to HF management, including therapeutic options [5,6]. Early palliative discussion with advance directive (AD) documentation is one approach to reduce patients’ and their caregivers’ burden [7]. Additionally, it can enhance patients’ autonomy by facilitating their active engagement in therapeutic and palliative discussion regarding prognosis, goals of care, symptom management, advantage/disadvantage of the therapeutic options, and other palliative concerns/issues, which facilitates informed decision-making [4,7]. Although the right time for palliative consultation is controversial, early integration of palliative care and/or AD documentation into standard care in the routine practice is highly recommended [6,8] which is more likely to improve the quality of one’s end-of-life (EoL) care aligned with personal values or treatment wishes, and timely consultation/or transition to palliative and hospice care, accordingly [7,9,10].

However, despite the proved benefits of palliative care and AD utilization in HF in Western countries [11], such as fewer resource utilization [12,13], AD is still underused even for patients with advanced HF receiving palliative care, and the documentation is suboptimal [14,15,16]. Cultural and ethnic disparities in suboptimal use of ADs were more substantial in the general and HF population. For example, black and/or Hispanic races compared with white race are less likely to have ADs [17,18]. In Korea, public attention to and awareness about ADs in the non-malignancy context have just emerged since the enforcement of the Act regarding EoL life-sustaining treatments (LSTs) (hereafter, the LST Act) [19]. Then, two legal forms of ADs are currently available: any adult who is 19 years and over can register one’s “Advance Directive on Life-sustaining Treatment” in a registry agency designated by the Ministry of Health and Welfare, while a person with a terminal condition or at the end stage of life can sign the “Life-Sustaining Treatment Plans” with one’s physician’s assistance [20]. Thus, ADs in Korean patients with HF have not been thoroughly examined yet.

To facilitate documentation of ADs, multi-faceted factors associated with completion of ADs need to be examined [21]. Some demographic and clinical factors, such as older age [14,22,23], female sex [14,23], higher education [22], certain comorbidities (cancer and renal dysfunction) [14], and worse HF progress [24] were reported to have their relationships to completion of ADs, AD attitudes, and/or EoL care more likely than each counterpart. Relatively, a few studies reported some modifiable factors, such as AD awareness, functional status, and depressive symptoms associated with their relationships to such EoL care outcomes. For example, awareness of AD was associated with more agreement in ADs between patients with cancer and their caregivers than no awareness of AD [25]. Better functional status in daily activities was associated with expressing EoL preference [23], but was not associated with having an AD in patients with HF [14]. Depression between HF patients expressed or not-expressed EoL preference, or depressive symptoms between those with or without an AD did not differ [14,23]. On the other hand, depressive symptoms were associated with having an AD in patients with cancer [26], or depressive symptoms between HF patients expressed or not-expressed EoL preference were different [23]. Based on the findings of the prior studies [14,23,24,25,26], we hypothesized that HF patients with poorer prognosis, lower awareness of ADs, worse functional status, and more severe depressive symptoms had higher likelihood of completing ADs compared to their counterparts. However, further research is needed because of some inconsistent findings in the literature and lack of examination related to AD completion factors. Therefore, the purpose of this study was to explore EoL values, treatment preference, and proxy of patients with HF using a Korean Advance Directive (K-AD) questionnaire and examine factors associated with the completion of the K-AD survey.

## 2. Materials and Methods

### 2.1. Study Design and Data Collection

Outpatients of a university-affiliated hospital (a convenience sample) were invited to participate in this descriptive, correlational study. The institutional review board of the hospital approved this study (GBIRB2017-058). Each patient signed a written informed consent statement prior to data collection. Then, a research coordinator who has expertise in hospice and palliative nursing care assisted patients with HF to complete the questionnaires to collect data on AD (EoL value, treatment directives, and proxy) and possible factors associated with completion of AD face to face during patients’ clinic visits for routine care.

### 2.2. Study Population

The eligibility criteria for this study were as follows: (a) age of 18 to 85 years, (b) documented diagnosis of HF with disease duration ≥ 6 months, and (c) optimal medical therapy, including beta-blockers, angiotensin-converting enzyme inhibitors or angiotensin receptor blockers, and diuretics. Patients were not eligible if they met at least one of the following criteria: (1) had end-stage HF that meets the hospice care qualification; (2) were candidates for heart transplantation or left ventricular assist device; (3) had a comorbid terminal illness that meets the requirements of impending hospice and palliative care [19], such as terminal cancer, acquired immune deficiency syndrome, chronic obstructive pulmonary disease, or chronic liver cirrhosis; and (4) had documented comorbidities that accompany serious cognitive impairment and prevent observation of patient autonomy for his or her medical care, such as dementia/Alzheimer’s disease and (traumatic) brain or psychiatric disorders.

### 2.3. Measures

#### 2.3.1. ADs

The K-AD survey questionnaire that was developed as a model for proposal of a culture-oriented, reality-based, and user-friendly AD in Korea was used to explore EOL values, treatment directives, and proxy designation [27]. After the enforced LST Act [19], the developer and co-authors revised the original version, with the four options of treatment directives for the non-malignancy and an additional option, “chemotherapy” for the malignancy. Using the K-AD, patients were asked to freely state personal values, determine their preference for each four treatment options (cardiopulmonary resuscitation [CPR], ventilation support, hemodialysis, and hospice care), and designate a proxy if available at the EoL moment. The K-AD was administered to explore wishes and preference for EOL care of Korean adults with cancer and those without cancer in clinical and nonclinical settings [27,28,29,30]. The validity of this instrument has been supported in cancer patients and their caregiver dyad by their concordance on the K-AD [28] and the significant associations of patients’ perceptions regarding AD and the K-AD treatment preferences in community-dwelling older adults with chronic diseases [30].

#### 2.3.2. HF Prognosis 

The Seattle Heart Failure Model (SHFM) was used to estimate 1-, 2-, and 5-year survival (mortality) and mean life expectancy of patients with HF with age ranging from 18 to 85 years [29]. The SHFM prognosis score was based on information about the following factors: New York Heart Association (NYHA) classes, ejection fraction, or systolic blood pressure, medications, diuretic use, laboratory data, and devices [29] that a cardiology physician provided after a patient’s clinic visit. SHFM scores were generated using the original published equation, with higher scores indicating poor HF prognosis [29].

#### 2.3.3. AD Awareness

Awareness of ADs was asked by a query if patients with HF were heard of what ADs were, or they had an experience in signing an AD.

#### 2.3.4. Functional Status in Daily Activities

The Korean Activity Status Index (KASI) was a measure of functional status in daily activities [30]. This scale consists of 15 daily physical activities with each activity having a weighted value depending on the required energy expenditure for performance. Possible scores range from 0 to 79, with higher scores indicating better functional status. Its psychometric properties were documented in patients with largely cardiovascular disease who underwent treadmill test [30].

#### 2.3.5. Depressive Symptoms

The Patient Health Questionnaire (PHQ) was a measure of depressive symptoms [31]. The PHQ is a 9-item questionnaire with each designing on a four-point Likert scale to evaluate the presence and severity of one’s depressive symptom experiences over the past two weeks. Possible scores range from 0 to 27, with higher scores indicating more severe depressive symptoms. The reliability and validity of this instrument have been supported in patients with HF [32].

#### 2.3.6. Demographic and Clinical Characteristics

The research coordinator collected demographic data, including age, sex, marital status, education, and awareness/need for ADs, using the standard form developed by the authors. A physician reviewed the medical records for clinical data on prescribed medications, etiology and duration of HF, and left ventricular ejection fraction (LVEF). The comorbidity index score was also calculated by the Charlson Comorbidity Index [33]. HF severity was assessed using the NYHA classification in which a physician classified an individual’s HF severity from class I (asymptomatic) to IV (unable to perform any physical activity without HF symptoms) based on symptom severity encountered during daily physical activities [34].

### 2.4. Statistical Analysis 

The Statistical Package for the Social Sciences (version 25.0; IBM, Armonk, NY, USA) [35] was used to analyze the data. The statistical significance level was set at a *p*-value < 0.05. Content analysis was performed to extract major themes of value statements. Frequencies for each of the four treatment directives were computed using descriptive statistics. Chi-square tests or t-tests were performed to compare patient demographic/clinical characteristics between patients who completed or did not complete the K-AD survey. Logistic regression analysis with Enter method was used to examine factors associated with the completion of the K-AD survey. ‘The completion of the K-AD survey’ was defined as responses to all or part of the three-component K-AD questionnaire, i.e., EoL values, treatment directives, or proxy designation. All variables of age, sex, education, caregiver, comorbidity, SHFM risk, AD awareness, functional status in daily activities, and depressive symptoms were entered into the model simultaneously.

## 3. Results

Sixty-seven patients with HF (mean age, 67.0 years (±11.8), 61.2% male) were included in the data analysis (Table 1). The mean education year was 8.7 (±4.5). More than half were married (64.2%). The mean comorbidity index score was 2.6 (±1.9). The mean HF duration was 52.9 months (±50.9). More than half had NYHA class II (58.2%). The mean LVEF was 35.9% (±9.4). The underlying diseases of HF were largely from ischemic cardiomyopathy (55.2%). HF prognosis ranged from a rounded integer of SHFM score to the nearest integer of between −1 and 2, with a mean SHFM score of 0.2 (±0.8; interquartile range, −0.4 to 0.8). Patients with HF demonstrated poor awareness about ADs, with only 13.4% reporting awareness or heard of ADs. 

### 3.1. Differences of Demographic and Clinical Characteristics between Patients Who Completed and Did Not Complete the K-AD Questionnaire

Of 67 patients, 35 (52.2%) completed all or part of the three-component K-AD questionnaire, and 32 patients (47.8%) did not state their EoL values, treatment directives, or surrogate decision-making person (Table 2). K-AD completers had a significantly higher education, better HF prognosis, more prescriptions of angiotensin converting enzyme inhibitors, and lower estimated five-year mortality than non-completers. The most highly valued EoL statement was comfortable death (*n* = 15), avoiding family burden (*n* = 6), or both (*n* = 3), followed by “do not have EoL values” (*n* = 9), and other responses (organ donation and do not know, each *n* = 1) (Table 3). Among four treatment directives, 68.6% of patients preferred hospice care preference; 28.6% (ventilation support and hemodialysis) to 42.9% (CPR) preferred aggressive treatment. Children (*n* = 13) and spouse (*n* = 10) were the most frequently named proxies. 

### 3.2. Factors Associated with the Completion of the K-AD Survey: Logistic Regression

Among age, sex, education level, existence of caregiver, comorbidity, HF prognosis, AD awareness, functional status in daily activities, and depressive symptoms, sex, HF prognosis, and functional status in daily activities were significantly associated with the completion of the K-AD survey. Female sex (Odds Ratio [OR] = 0.167, *p* = 0.027), poorer HF prognosis (OR = 0.156, *p* = 0.006), and better functional status in daily activities (OR = 0.905, *p* = 0.001) were associated with less likelihood of completing the K-AD survey (Table 4).

## 4. Discussion

This study initially reports EOL values, treatment directives, and proxy designation in patients with HF. The findings of this study are valuable regarding ADs in patients with HF, including low awareness of ADs and the needs for more attention to HF patients who are female, have poor prognosis, and better functional status. In this study, patients with HF seemed to have positive attitudes toward ADs because approximately half of the patients with HF completed the AD questionnaire, even though the majority (86.6%) of patients reported a lack of awareness of the ADs. On the basis of responses of patients with HF to the AD survey, their attitudes appeared to be favorable, particularly toward comfort care at the EoL moment because they put the highest values to comfort death (42.8%), avoiding family burdens (17.1%), or both (8.6%). Further, approximately two-thirds (68.6%) had a high preference for hospice care, while one-third or more had preferences for aggressive treatments, ranging from 28.6% (ventilation support and hemodialysis) to 42.9% (CPR). These results were somewhat similar, but also somewhat different from the findings in prior studies depending on the healthcare contexts [27,28,36]. Compared to patients with HF, chronically ill elderly people in the community demonstrated a relatively lower value for comfortable death (35%) [27], whereas in the clinical setting, patients with malignancies had a relatively higher value for comfortable death (73.8%) and comfortable death with absence of pain (47.4%) [28]. For advance treatment options, 79.5% of patients with cancer [28] and 56.4% of older adults in the community [27] desired hospice care, which are similar to the finding in this study. In contrast, 28.6% to 42.9% of patients with HF preferred aggressive treatment in this study, while only 20.5% of patients with cancer [28] and 23.3% to 24.0% of older adults in the community preferred aggressive treatment, such as CPR and ventilation support [27]. In patients with hematological disorders [36], even fewer patients preferred CPR (8.6%) and ventilation support (5.7%). These findings suggest that more in-depth discussion regarding ADs is needed for patients with HF to prevent unnecessary aggressive treatment and to reduce patient burden due to the unnecessary aggressive treatment at the EoL.

Regarding the proxy appointment, patients with HF preferred to designate their children as a surrogate decision-maker most frequently in case of their loss of decisional capacity at the EoL (37.1%), followed by spouses (28.6%). Similar to patients with HF, community-dwelling older persons (the mean age, 77 years) were likely to appoint their children as a proxy predominantly (77.1%), followed by spouses (17.5%) [29]. Somewhat different results of proxy appointments were reported by patients with cancer (the mean age, 58.43 years), with spouses being appointed as a proxy most frequently (70.5%), followed by adult children (20.4%) [28]. These findings suggest that family members [11], particularly spouses and children who were primarily designated as a proxy regardless of illnesses, are encouraged to participate in an in-depth discussion regarding ADs to understand and be informed of patients’ EoL values.

Under the circumstance of the LST Act in execution nationwide for individuals with (LST plans) or without terminal conditions (ADs on LSTs) [19,37], it is valuable to know that more than half (52.2%) of patients with HF in this study responded to all or part of the K-AD questionnaire regarding EoL values, treatment options, and/or surrogate decision makers. ADs have been exclusively targeting the population with cancer. However, considering the symptom burdens of patients with symptomatic HF, needs for palliative care in patients with HF are similar to those in patients with advanced cancer [38]. In addition, over the past decades, attention to AD utilization in HF has been growing; further, its early integration as part of the standard care is highly recommended [6,9,39]. However, inadequate or delayed use of palliative consultation and/or ADs in patients with HF was often reported [7,14,15]. Thus, it is important to know factors associated with utilization of ADs to facilitate the utilization.

In this study, one demographic characteristic (female sex) was associated with less likelihood of completing AD documentation. In a prior study that was conducted in the U.S. [14], sex was not a factor associated with completing AD in an adjusted model, but in an unadjusted model, female patients with HF were more likely to complete AD than male sex. On the other hand, in another U.S. study [15], in an adjusted model, female sex was associated with more likelihood of having documented AD than male sex. In all the current and prior studies, sex ratios were similar. The reasons for the inconsistent findings in the relationships between sex and completion of AD may be, in part, due to differences in cultures and also the independent variables in the models. In Asian cultures, including Korean culture, patients want family members to make EoL decisions [40]. Thus, more encouragement for females to participate in EoL and AD discussion may be beneficial in Korean culture. On the other hand, the prior studies did not include functional status in daily activities and HF progress as the independent variables, which showed significant associations with completion of AD in this study. Thus, further studies are needed to examine the relationship between sex and completion of ADs more thoroughly.

In this study, the majority of patients with HF (87%) did not aware ADs, which is consistent with the finding in a study of a community-dwelling older Korean people with chronic diseases (91%) [30]. We selected awareness of ADs as a possible factor associated with the completion of the K-AD. However, AD awareness was not significantly associated with the K-AD completion in this study. The non-awareness of ADs in the majority of patients with HF may be one reason for this lack of a significant relationship given the lack of variability in knowledge. In a pilot study, patients with HF reported knowledge deficit as a major reason for incomplete K-AD survey [41]. Thus, further research is warranted to examine this relationship in larger sample studies with more variability in knowledge. In addition, interventions need to be developed and delivered to this population to improve knowledge on ADs to facilitate more active discussion regarding ADs in patients with HF and to help them make more appropriate and reasonable treatment decision considering their preferences.

Additional factors associated with the less likelihood of completing the AD survey were poorer HF prognosis and better functional status in daily activities. In a prior study [24], HF prognosis (estimated one-year mortality) was also associated with having AD or no preference of resuscitation. In prior studies, the relationships between functional status in daily activities and AD or EoL preference were inconsistent. Better functional status in daily activities was not associated with an AD in patients with HF [14], but was associated with expressing EoL preferences [23]. However, the significant association in the latter study was based on bivariate analysis. Thus, further studies are needed to examine the relationship between functional status and completion of or having ADs. However, the significant associations of poorer HF progress and better functional status with the completion of the AD survey in this study imply the needs for early discussion of AD and advance care planning (ACP) in patients with HF. If early discussion is started when patients’ functional status is still good, it may facilitate AD completion before the overall HF progress becomes poor. This is also important considering the prognostic uncertainty in patients with HF.

Prognostic uncertainty is a major challenge for the initiation of AD and/or ACP discussion, which precludes optimal AD utilization [6,7,42]. Due to the prognostic uncertainty of HF, palliative care needs of patients with HF is more likely to be better addressed using a HF-focused palliative care model [39]. The significant relationship between HF prognosis and completion of AD in this study implies that prognostic consultation can start as early as possible with integration into the routine standard care for those with better HF prognosis. This is important because no AD commonly leads to delayed EoL care and care that does not match with patients’ treatment wishes. For example, in patients with relatively poor HF progress (NYHA classes III/IV, 87%; SHFM one-year mortality, 29%; median life expectancy, 2.8 years), the survival period since palliative consultation was less than 1 month (median duration = 21 days), implying that such recommended care was yet delayed in those with advanced HF near death [42]. Patients with advanced HF who did not possess an AD were more likely to receive aggressive care, such as ventilation support or admission to the intensive care unit, during their EOL stage [14], while elderly Americans who had documented preferences on an AD were more likely to receive care as indicated by their wishes than those without ADs [43].

This study has some limitations. Given a recent LST Act in execution and increased attention to the ADs in non-malignancy contexts beyond malignancy in Korea [19], development of an organizational policy and system for AD documentation is currently in progress. Thus, AD treatment preference using a K-AD questionnaire may not reflect actual desires for AD documentation and require validation of the prevalence of AD utilization in a representative sample. The prognostic importance of ACP in patients with HF is supported by this study, while empirical evidence is still lacking in its concept and awareness, further studies in the investigation of these associations with planning ACP and/or ADs are needed. Additionally, this study supports early adoption of the AD based on more patients with HF showing that patients with poor HF prognosis had less likelihood of completing the AD survey, but the right time to begin such care is unclear. Thus, studies are warranted to determine the most appropriate time to start prognostic communication of AD as part of ACP using a meticulous study design and model of care and, further, proving its advantages through various patient outcomes. The sample of this study was enrolled from one medical center, which can limit the generalizability of the findings. Lastly, the AD instrument has not been thoroughly tested in patients with HF. Thus, the psychometric properties of the instrument and the relationships examined in this study need to be examined in larger samples from several recruitment sites.

The results of this study provide important implications that can test the feasibility and needs for palliative care and AD use in the Korean population. It seems feasible to introduce palliative care in HF across the stages, in which such an unfamiliar and difficult topic was well accepted in Korean patients with HF. Cultural and ethnic disparities in suboptimal use of ADs were more substantial in the general and HF population. For example, black and/or Hispanic races compared with white race are less likely to have ADs. Under the cultural and/or ethnical influences on ACP and AD use [17,18], the results of this study extended patient data to East Asian cultural tradition, particularly under Confucian beliefs where ACP and ADs in non-malignancy contexts began [44,45]. These findings reveal that early prognostic communication during the routine care is possible for those with better prognosis that is fundamental in ACP particularly at the beginning of the diagnosis of HF. Thus, future research should investigate whether early prognostic discussion as part of the ACP with integration into the standard care of HF facilitates AD documentation. Research studies are also needed to generate ample evidence regarding palliative consultation in HF and its efficacy in accomplishing a wide spectrum of better patient outcomes, in which these provisional efforts could be based on grounds for institutional rules and regulations and policy-making in the palliative care of patients with HF. Especially, consultation of ACP and AD use needs to be provided to HF patients who are female, have poor prognosis, and better functional status because they showed less likelihood of completion of the K-AD survey.

## 5. Conclusions

In conclusion, more than 50% completion rate of AD in this study implies that patients with HF seemed to have positive attitudes toward ADs. Although the majority of the completers preferred hospice care, still considerable portions of the completers preferred aggressive treatments, such as CPR or ventilation support. The findings of this study suggest some targets of interventions to facilitate completion of AD, such as female sex, poorer HF prognosis, and better functional status in daily activities. These findings suggest that early, in-depth AD discussion as part of the ACP needs to be started earlier in patients with HF to facilitate completion of AD and provision of necessary and value-matched EoL care.

## Figures and Tables

**Table 1 ijerph-18-01780-t001:** Demographic and clinical characteristics of patients with heart failure (*n* = 67).

Variables		*n* (%) or Mean ± SD	Range
Age, years		67.0 ± 11.8	37–85
Education, years		8.7 ± 4.5	0.0–16.0
Sex	Male	41 (61.2)	
Marital status	Married	43 (64.2)	
Employment	Employed	20 (29.9)	
Caregiver	Yes	52 (77.6)	
Comorbidity		2.6 ± 1.9	1–10
Heart failure duration, months		52.9 ± 50.9	6–201
Left ventricular ejection fraction, %		35.9 ± 9.4	17.0–67.0
NYHA classes	I	8 (11.9)	
	II	39 (58.2)	
	III	20 (29.9)	
	IV	0 (0.0)	
Etiology	ICM	37 (55.2)	
	DCM	15 (22.4)	
	HTN	5 (7.5)	
	VHD	4 (6.0)	
	AFib	3 (4.5)	
	Alcoholic	3 (4.5)	
Medication, yes	ACE inhibitor	32 (47.8)	
	ARB	21 (31.3)	
	Beta-blockers	57 (85.1)	
	Statin	44 (65.7)	
	Aldosterone blocker	17 (25.4)	
	* Diuretics	35 (52.2)	
AD awareness (yes)		9 (13.4)	
** SHFM risk score		0.2 ± 0.8	−1 to 2
1-year mortality		6.6 ± 5.4	0.9–23.0
2-year mortality		12.5 ± 9.7	1.8–42.0
5-year mortality		29.4 ± 19.0	4.9–78.0
Life expectancy		10.5 ± 4.5	3.2–23.2

ACE, angiotensin-converting enzyme; AD, advance directives; AFib, atrial fibrillation; ARB, angiotensin receptor blocker; DCM, dilated cardiomyopathy; HTN, hypertension; ICM, ischemic cardiomyopathy; NYHA, New York Heart Association; SD, standard deviation; SHFM, Seattle Heart Failure Model; VHD, valvular heart disease. * Includes furosemide, torsemide, and hydrochlorothiazide. ** Calculated with the equation of natural log(natural log(SHFM estimated 1-year survival)/natural log(0.9604)).

**Table 2 ijerph-18-01780-t002:** Differences of demographic and clinical characteristics between completers and non-completers of the Korean advance directives questionnaire.

Characteristics	Korean Advance Directive Model		*t* or *χ*^2^	*p*
	Non-Completers (*n* = 32) *n* (%)/Mean ± SD	Completers * (*n* = 35) *n* (%)/Mean ± SD		
Age, years	69.3 ± 10.8	64.8 ± 12.5	1.55	0.125
Education, years	7.7 ± 4.0	10.2 ± 4.6	−2.86	0.006
Sex, male	17 (53.1)	24 (68.6)	1.68	0.195
Marital status, married	17 (53.1)	26 (74.3)	3.26	0.071
Employment, employed	9 (28.1)	11 (31.4)	0.09	0.768
Caregiver, yes	27 (84.4)	25 (71.4)	1.61	0.204
Comorbidity	2.8 ± 2.3	2.4 ± 1.5	0.96	0.343
NYHA (III)	12 (37.5)	8 (22.9)	1.71	0.191
LVEF	34.8 ± 8.9	36.8 ± 9.9	−0.86	0.394
Heart failure duration, months	51.4 ± 48.6	54.2 ± 53.5	−0.22	0.826
Heart failure etiology, ischemic	16 (50.0)	21 (60.0)	0.68	0.411
ACE inhibitor, prescription	11 (34.4)	21 (60.0)	4.40	0.36
ARB, prescription	11 (34.4)	10 (28.6)	0.26	0.609
Beta-blockers, prescription	26 (81.3)	30 (85.7)	0.24	0.622
Statin, prescription	21 (65.6)	23 (65.7)	<0.01	0.994
Aldosterone blocker, prescription	9 (28.1)	8 (22.9)	0.24	0.621
Diuretics, prescription **	19 (59.4)	18 (51.4)	0.43	0.514
AD awareness	3 (9.4)	6 (17.1)		0.480 ***
SHFM risk score ****	0.4 ± 0.8	<0.1 ± 0.8	2.04	0.046
1-year mortality	7.7 ± 5.8	5.5 ± 4.8	1.74	0.086
2-year mortality	14.7 ± 10.3	10.6 ± 8.8	1.75	0.085
5-year mortality	33.9 ± 19.8	25.4 ± 17.7	1.86	0.067
Life expectancy	9.3 ± 3.9	11.6 ± 4.7	−2.18	0.033

ACE, angiotensin-converting enzyme; AD, advance directive; comorbidity, Charlson Comorbidity Index score (range: 1.0–10.0); LVEF, left ventricular ejection fraction (range: 17.0–67.0%); NYHA, New York Heart Association. NYHA functional class I/II vs. III. No patients were at NYHA functional class IV. * Patients who completed all or part of the 3-component Korean Advance Directive model; ** Diuretics includes furosemide, torsemide, and hydrochlorothiazide; *** Fisher’s exact test; **** Calculated with the equation of Ln(Ln(SHFM estimated 1-year survival)/Ln(0.9604)).

**Table 3 ijerph-18-01780-t003:** End-of-Life statements and proxy designation among completers of the K-AD model (*n* = 35).

End-of-Life Statements * (*n* = 25)	*n* (%)	Proxy Designation ** (*n* = 26)	*n* (%)
Want to die comfortable	15 (42.9)	Children	13 (37.1)
Avoid family burden	6 (17.1)	Spouse	10 (28.6)
Dying comfortably without burden on family	3 (8.6)	Siblings	3 (8.6)
Do not have end-of-life values	9 (25.7)	Relatives	1 (2.9)
Donate organs	1 (2.9)		
Do not know	1 (2.9)		

* Twenty-five (71.4%) provided multiple responses to end-of-life value statement. ** Twenty-seven (77.1%) provided proxy information.

**Table 4 ijerph-18-01780-t004:** Correlates of completion of the Korean advance directives questionnaire.

Factors	B	*p*	OR	95% CI
Age	−0.040	0.363	0.961	0.882, 1.047
Sex	−1.791	0.027	0.167	0.034, 0.813
Education, years	0.121	0.142	1.128	0.960, 1.325
Caregiver	−1.374	0.112	0.253	0.047, 1.377
Comorbidity	−0.170	0.341	0.844	0.595, 1.196
SHFM risk	−1.855	0.006	0.156	0.042, 0.582
AD awareness	0.564	0.550	1.758	0.276, 11.190
* Functional status	−0.099	0.001	0.905	0.852, 0.0962
Depressive symptoms	0.035	0.548	1.035	0.925, 1.159
Model summary	Chi-square: 32.261, *p* < 0.001. Nagelkerke R^2^ = 0.510			

Abbreviations: AD, advance directive; CI, confidence interval; HF, heart failure; OR, odds ratio; SHFM, Seattle Heart Failure Model. * Functional status in daily activities was assessed by the Korean-Activity Status Index.

## Data Availability

The data presented in this study are available on request from the corresponding author. The data are not publicly available due to ethical issue.

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
