# Peer review of "Advance Directives and Factors Associated with the Completion in Patients with Heart Failure"

_ijerph, 2021, doi:10.3390/ijerph18041780_

Round 1
Reviewer 1 Report
In general the article is well written. It has a clear objective, uses an appropriate method to achieve the objective, the results are well presented, and the discussion is interesting. However, the study has important limitations. Some cannot be solved (such as the sample size) but must be specified as study limitations, while other deficiencies can be resolved.
These limitations are further specified below:
- It is not specified whether the K-AD survey questionnaire is a validated questionnaire.
- It would be recommendable to attach the K-AD survey questionnaire.
- The purpose of this study was to explore EoL values, treatment preference, and proxy of patients with HF using the K-AD questionnaire and examine factors associated with the completion of the K-AD survey. However, in the results is not presented (nor discussed) the patients' proxies.
- It cannot be said that “Comfortable death” was highly valued, with N = 15. It can be said that it was more evaluated than other values, but highly valued?
- Although ADs in Korean patients with HF have not been thoroughly examined yet, the sample of patients is very small, especially if other studies conducted on ADs in HF in other countries or in other pathologies are reviewed. Even more so considering that only approximately half of the patients with HF completed the AD questionnaire. All of this makes it difficult to state statements and draw valid conclusions, although there are certain statistical differences. All this should be commented / explained in the discussion as an important limitation of the study, pointing out that this study is the beginning of a line of research, but that studies with larger samples are needed.
- Another important limitation of the study, and that should be pointed out in the discussion, is that it is a study carried out in a single center, which limits the external validity of it results. Therefore, to draw extrapolable conclusions, are need: multicenter studies with larger samples.
- In the discussion nothing is said about the low knowledge of ADs among patients, and what can be done about it.
- The bibliography must be updated, because in the last four years a lot has been published about ADs in HF.
Author Response
Dear Reviewer:
On behalf of my coauthors, I thank the reviewers for their comments and recommendations. We have made the following changes to the manuscript to address their concerns. We have also revised the manuscript while paying particular attention to the language, with editing support from a scientific editing company. The revisions in the manuscript text are highlighted in yellow as well as using the Track Changes function of MS Word.
Please see the attachment.
We thank you again for your recommendations. We believe they have strengthened our manuscript significantly.
Sincerely,
Authors

Reviewer 2 Report
please see attached file

Author Response

(The authors gave the same response as above.)

Round 2
Reviewer 1 Report
With the changes made, the article could be published. I advise that the English and the bibliography be checked.
Author Response
Dear Reviewer:
On behalf of my coauthors, I thank the reviewer for their comments and recommendations. We have made the following changes to the manuscript to address their concerns. The revisions in the manuscript text are highlighted in skyblue. Please see the attachment.
Sincerely,
Authors

Reviewer 2 Report
I appreciate the authors effort to revise the paper and all my comments were adequately addressed.
Author Response
Dear Reviewer:
We thank you again for your recommendations. We believe they have strengthened our manuscript significantly.
Sincerely,
Authors